# Magnetization reversal driven by low dimensional chaos in a nanoscale ferromagnet

Eric Arturo Montoya [1], Salvatore Perna[2], Yu-Jin Chen [1], Jordan A. Katine[3], Massimiliano d'Aquino [4], Claudio Serpico[2] & Ilya N. Krivorotov [1]

Energy-efficient switching of magnetization is a central problem in nonvolatile magnetic storage and magnetic neuromorphic computing. In the past two decades, several efficient methods of magnetic switching were demonstrated including spin torque, magneto-electric, and microwave-assisted switching mechanisms. Here we experimentally show that low-dimensional magnetic chaos induced by alternating spin torque can strongly increase the rate of thermally-activated magnetic switching in a nanoscale ferromagnet. This mechanism exhibits a well-pronounced threshold character in spin torque amplitude and its efficiency increases with decreasing spin torque frequency. We present analytical and numerical calculations that quantitatively explain these experimental findings and reveal the key role played by low-dimensional magnetic chaos near saddle equilibria in enhancement of the switching rate. Our work unveils an important interplay between chaos and stochasticity in the energy assisted switching of magnetic nanosystems and paves the way towards improved energy efficiency of spin torque memory and logic.

[1] Department of Physics and Astronomy, University of California, Irvine, CA 92697, USA. [2] Department of Electrical Engineering and Information Technology, University of Naples Federico II, 80125 Naples, Italy. [3] Western Digital, 5600 Great Oaks Parkway, San Jose, CA 95119, USA. [4] Engineering Department, University of Naples "Parthenope", 80143 Naples, Italy. Correspondence and requests for materials should be addressed to I.N.K. (email: ilya.krivorotov@uci.edu)

The striking complexity that may arise in the trajectories of a nonlinear deterministic dynamical system was discovered by Henri Poincaré in the 1880s while studying the three-body problem of celestial mechanics[1]. This pioneering work demonstrated strong sensitivity of the dynamic trajectories to small perturbations and gave birth to a branch of science that studies chaos—deterministic dynamics extremely sensitive to initial conditions[2]. The ideas of Poincaré led to the development of Kolmogorov–Arnold–Moser (KAM) theory[3], which describes the emergence of chaotic dynamics arising from perturbations applied to integrable Hamiltonian systems. It is now well established that chaotic dynamics is ubiquitous—it is encountered in celestial mechanics[4], biology[5], fluid dynamics[6], astronomy[7], as well as mechanical and telecommunications engineering[8]. Notably, fluid turbulence—the central problem in aerospace engineering—can be viewed as a manifestation of chaotic dynamics[9]. From the fundamental point of view, the chaotic nature of molecular dynamics has played a key role in establishing rigorous foundations of statistical mechanics in connection with the ergodic hypothesis and the law of increase of entropy[10].

Remarkably, chaos may already arise in dynamical systems with a few degrees of freedom, such as systems described by three state variables or by two state variables in the presence of a time-varying external excitation[11]. These low-dimensional dynamical systems are particularly important for studies of chaos because time evolution of all state variables can be traceable in both experiments and numerical simulations performed for such testbed systems[12].

In the field of magnetism, chaotic dynamics was previously observed in ferromagnetic resonance (FMR) experiments at high excitation power[13]. In FMR measurements, magnetization dynamics is excited by a microwave frequency ac magnetic field applied to a macroscopic ferromagnetic body[14]. At low ac power levels, only the spatially uniform mode of the magnetic precession is excited resulting in periodic motion of the magnetization at the frequency of the ac drive. When ac power increases above a threshold value, nonlinear coupling of the uniform mode to a continuum of spatially non-uniform spin wave modes gives rise to an exponential growth of the amplitude of multiple modes[15]. The resulting dynamic state of magnetization is a continuum of interacting large-amplitude spin waves that can exhibit quasi-periodic, chaotic, and turbulent types of dynamics[13,16]. Such nonlinear magnetization dynamics is currently a very active area of study[17,18].

While much work was done towards understanding of chaotic dynamics in magnetic systems with continuous degrees of freedom[13], experimental studies of chaos in magnetic systems with a few degrees of freedom are lacking. In this article, we experimentally and theoretically investigate chaotic dynamics in a ferromagnetic system with two degrees of freedom subject to a periodic external drive. This low-dimensional magnetic chaos is achieved in a magnetic nanoparticle driven by alternating spin transfer torque.

Geometric confinement discretizes the spectrum of spin wave eigenmodes in a nanomagnet and thereby suppresses energy- and momentum-conserving nonlinear spin wave interactions present in bulk ferromagnets with continuous spin wave spectrum[19]. This suppression of nonlinear spin wave interactions allows for excitation of large-amplitude quasi-uniform precession of magnetization without simultaneous excitation of other spin wave modes of the system[20,21]. We demonstrate that this type of magnetic dynamics specific to nanoscale ferromagnets provides a perfect testbed for studies of low-dimensional magnetic chaos[22–24].

Our studies reveal that chaotic magnetization dynamics induced by alternating spin torque has a profound effect on thermally-assisted switching of magnetization in a nanomagnet.

This intriguing coupling between low-dimensional deterministic chaos and temperature-induced stochastic dynamics[25–27] is not only of fundamental interest but also of significant practical importance. Indeed, non-volatile magnetic storage technologies such as spin transfer torque memory (STT-RAM)[28,29] and microwave-assisted magnetic recording (MAMR)[30–32] rely on thermally assisted switching of nanoscale ferromagnets. Additionally, innovative computing schemes, such as neuromorphic computing[33,34] and invertible logic[35], have been proposed in such systems in the telegraphic switching regime. Our work reveals that low-dimensional deterministic chaos can be employed for reduction of the effective magnetic energy barrier for switching of magnetization in a nanoscale ferromagnet and thereby paves the way towards more energy-efficient nonvolatile magnetic storage and logic technologies[36–38].

## Results

**Low-dimensional chaos in a nanomagnet**. In this article, we present experimental and theoretical studies of low-dimensional chaos in a nanoscale ferromagnet with biaxial magnetic anisotropy. The nanomagnet is a 1.8 nm thick $Co_{60}Fe_{20}B_{20}$ elliptical thin-film element with lateral dimensions of $50 \times 75$ nm$^2$ that is sufficiently small to support a single-domain ground state. We detect the direction of the nanomagnet magnetization electrically via embedding the nanomagnet as a free layer into a nanoscale magnetic tunnel junction (MTJ) illustrated in Fig. 1a. Rotation of the free layer magnetization with respect to the synthetic antiferromagnet (SAF) reference layers results in variation of the MTJ resistance via tunneling magneto-resistance (TMR) effect[39]. Since directions of the magnetic moments within the SAF are fixed[39], variation of the MTJ resistance with time arises solely from magnetization dynamics of the free layer.

The direction of the free layer magnetization can be described by a vector $m$ on a unit-sphere as shown in Fig. 1b. The single-domain free layer magnetization can therefore be described by two state variables, for example, $z$-component of $m$ and azimuthal angle $\phi$ in the $xy$ plane. Dipolar interactions give rise to magnetic shape anisotropy of the nanomagnet that is predominantly easy-plane with its hard axis along the film normal[40]. A weaker easy axis anisotropy is present in the sample plane with its easy axis parallel to the long axis of the ellipse. The biaxial anisotropy energy landscape of this system with two degrees of freedom can be visualized by drawing constant-energy contours on the unit sphere (Fig. 1b). This landscape consists of two magnetic potential energy wells near the energy minima at $m_x = \pm 1$ and two saddle points at $m_y = \pm 1$. The constant energy contours connecting the saddle points (thick black lines in Fig. 1b) are the heteroclinic orbits, which define the separatrices forming the boundaries of the potential wells.

In the following sections, we show that chaotic magnetization dynamics of the free layer nanomagnet can be induced by ac spin torque when $m$ passes sufficiently close to the separatrices (defined by unperturbed case). Specifically, chaotic dynamics is realized within a band of anisotropy energies around the separatrix energy (dark gray band in Fig. 1c). The width of this band of chaotic dynamics increases with increasing amplitude of the ac drive[41,42] after suitable threshold amplitude is exceeded. Infinitesimal changes of the initial direction of $m$ within this energy band are predicted to result in strong variation of the magnetization trajectory, which is the main signature of deterministic chaos.

The process of magnetization switching from one potential well into the other necessarily involves crossing the separatrices and, therefore, must proceed via the band of chaotic dynamics induced by the ac drive. Therefore, we expect the ac-driven chaotic

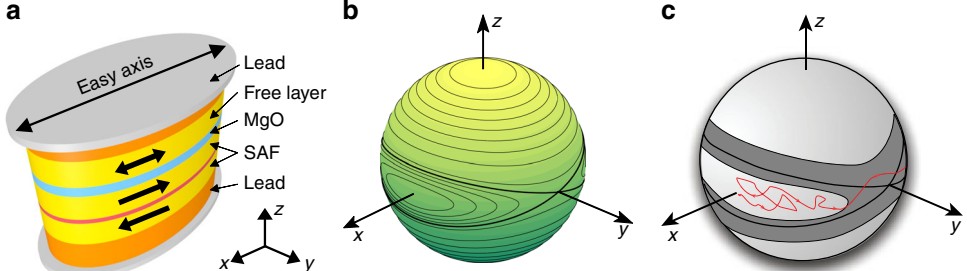

**Fig. 1** Device and free layer energy landscape. **a** Schematics of nanoscale magnetic tunnel junction consisting of a synthetic antiferromagnet (SAF) reference and a superparamagnetic $Co_{60}Fe_{20}B_{20}$ free layer separated by an MgO tunnel barrier. **b** Unit sphere representing the free layer magnetization direction $\boldsymbol{m}$ with contour lines of constant magnetic anisotropy energy. The biaxial magnetic anisotropy energy landscape consists of two energy minima at $\boldsymbol{m}$ parallel to the $x$-axis, two energy maxima at $\boldsymbol{m}$ parallel to the $z$-axis, and two saddle points for $\boldsymbol{m}$ parallel to the $y$-axis. **c** Schematic illustration of the effect of ac spin torque on thermally activated switching of the free layer. The free layer nanomagnet switching trajectory $\boldsymbol{m}(t)$ must pass through the dark gray band induced by ac spin torque where deterministic magnetization trajectories connect the two potential wells. The presence of this band of chaotic dynamics results in erosion of the boundary between the two potential wells and reduction of the effective energy barrier for thermally activated switching of magnetization between the wells

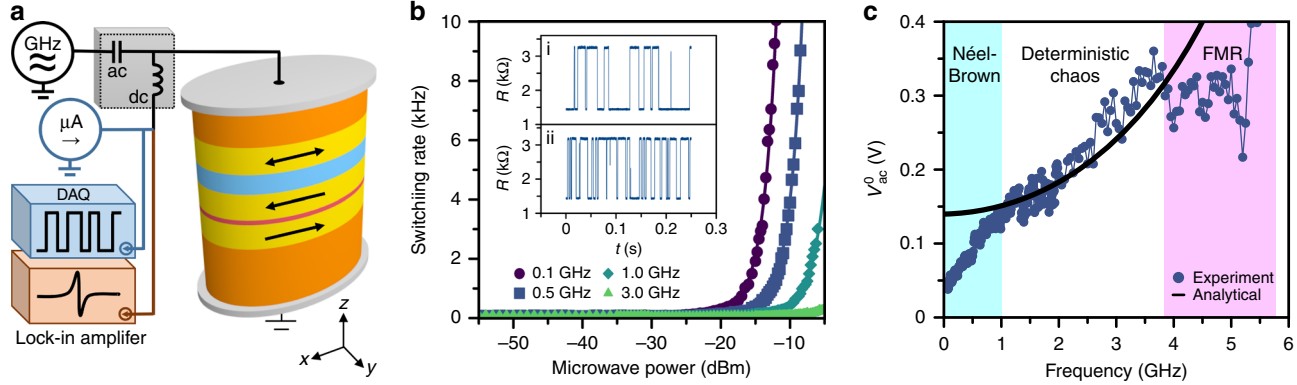

**Fig. 2** Random telegraph noise measurements. **a** Schematics for random telegraph noise (blue) and spin torque ferromagnetic resonance (brown) experiments. **b** Dependence of the free layer switching rate $w$ on applied microwave ac power and frequency. The solid lines are guides to the eye. The insets show resistance of the MTJ as a function of time measured at two values of a 0.5 GHz ac power: (i) $P_{ac} = -55$ dBm and (ii) $P_{ac} = -18.5$ dBm. **c** Threshold ac drive voltage $V_{ac}^0$ as a function of the ac drive frequency. The black solid line is theoretical prediction for the onset of ac-driven chaotic dynamics evaluated analytically at zero temperature

dynamics excited above appropriate threshold amplitude to affect the nanomagnet switching behavior. In this article, we experimentally investigate thermally activated switching between the potential wells schematically illustrated by a stochastic trajectory in Fig. 1c (red line). We study the effect of ac spin torque drive on the rate of thermally activated switching of the free layer nanomagnet and thereby probe the effect of chaotic dynamics on the switching process.

**Magnetization switching measurements**. In order to accelerate measurements of thermally-activated switching of the free layer nanomagnet, we employ MTJ samples with superparamagnetic free layers[43], in which the free layer stochastically switches between the two anisotropy energy wells at the rate of several tens of Hz. The superparamagnetic state is achieved via patterning the MTJ into a low-aspect-ratio ellipse with small volume, such that the in-plane shape anisotropy defining the energy barrier for switching is not too large compared to the thermal energy[43]. This system exhibiting random telegraph noise (RTN)[25,44] allows us to collect statistically accurate data on thermally activated switching rates and their modification by ac spin torque over experimentally convenient time of several hours at room temperature ($T = 300$ K).

The high value of TMR of the MTJ spin-valve allows us to monitor the RTN dynamics of the free layer in real time. The experimental setup for the RTN measurements is shown in Fig. 2a. A low-level probe current ($-25\,\mu$A) is applied to the MTJ and the voltage across the device is measured by a high-performance data acquisition system (DAQ) in real time (Methods). In these measurements, we apply a small in-plane magnetic field (3.7 mT) along the nanomagnet easy axis that compensates the stray field from the SAF layer acting onto the free layer and balances the dwell times of the free layer in the high-resistance (antiparallel, AP) and low-resistance (parallel, P) states. A microwave frequency ac voltage applied to the MTJ via the ac port of the bias tee gives rise to an ac spin torque applied to the free layer by spin-polarized electric current from the SAF layer. The switching rate of the free layer nanomagnet is the inverse of the dwell time $w \equiv 1/\tau$. Examples of time-domain RTN data are shown in the insets of Fig. 2b.

Example of the measured switching rate dependence on the applied microwave power $P_{ac}$ is shown in Fig. 2b for the ac spin torque frequencies $f = 0.1$, 0.5, 1.0, and 3.0 GHz. All these frequencies lie below the FMR frequency of the free layer $f_{FMR} = 5.1$ GHz (Supplementary Note 1 and Supplementary Figure 1), as determined from field-modulated spin torque ferromagnetic resonance (ST-FMR) measurements (Methods)[45]. Additionally,

all these frequencies are well above the low-frequency regime (few hundred Hz and below) where stochastic resonance effects[46] are observed (Supplementary Note 2 and Supplementary Figures 2 and 3), showing the observed effect does not arise from stochastic resonance.

The RTN data in Fig. 2b reveal that the switching rate of the free layer is strongly affected by ac spin toque with frequencies well below the FMR frequency of the free layer. Furthermore, lower frequencies of the ac drive have a stronger effect on the switching rate. These data clearly show that the observed effect of the ac drive on switching is not connected to the resonant excitation of the free layer magnetization (FMR). We argue that the observed effect of the sub-FMR-frequency ac drive arises from the low-dimensional chaotic dynamics induced by the ac spin torque.

The data in Fig. 2b clearly show that the effect of ac spin torque on the free layer switching has a threshold character in the ac power. These data also reveal that the threshold power decreases with decreasing frequency of the ac drive. In order to quantify the dependence of the threshold power on the ac drive frequency, we define the threshold power $P_{ac}^0$ as the ac power at which the switching rate $w$ doubles compared to its value in the absence of the ac drive.

Figure 2c shows frequency dependence of the ac threshold voltage $V_{ac}^0$ applied to the sample calculated from the measured threshold power $P_{ac}$ by correcting for frequency-dependent attenuation in the measurement circuit and impedance-dependent ac signal reflection from the sample[47]. The black solid line in Fig. 2c shows our zero-temperature theoretical prediction (discussed in the next section) for frequency dependence of the ac threshold voltage for the onset of chaotic magnetization dynamics. The prediction is in good agreement with our experimental data in a wide range of ac frequencies (1–4 GHz) below the FMR frequency. The deviations from the analytic prediction at higher and lower frequencies will be addressed in the Discussion section.

**Theoretical results**. From a theoretical point of view, the chaotic magnetization dynamics induced by ac torque can be studied with tools of nonlinear dynamical systems such as the Poincaré map[41]. Such analysis allows the definition of the concept of erosion of the stability region of magnetic equilibria by the ac drive[42], which provides a natural connection between deterministic chaos and thermally-activated switching over the energy barrier.

Magnetization dynamics for a uniformly magnetized particle driven by spin torque is described by the stochastic Landau–Lifshitz equation[48,49]:

$$\frac{d\boldsymbol{m}}{dt} = -\boldsymbol{m} \times \left( \boldsymbol{h}_{\text{eff}} + \alpha(\boldsymbol{m} \times \boldsymbol{h}_{\text{eff}}) - \beta(\boldsymbol{m} \times \boldsymbol{e}_{\text{p}}) + \nu \boldsymbol{h}_{\text{N}} \right), \quad (1)$$

where $\boldsymbol{m}$ is the magnetization vector of unit length $|\boldsymbol{m}| = 1$ (normalized by the saturation magnetization $M_s$), time is measured in units of $(\gamma M_s)^{-1}$ ($\gamma$ is the absolute value of the gyromagnetic ratio), $\boldsymbol{h}_{\text{eff}} = -\partial g/\partial \boldsymbol{m}$ is the effective field, $g = g(\boldsymbol{m}, \boldsymbol{h}_a)$ is the magnetic free energy, $\boldsymbol{h}_a$ is the external magnetic field, $\alpha$ is the damping constant, $\beta(t) = \beta_{ac} \cos(\omega t)$ is the normalized ac Slonczewski spin torque that can transfer energy to the nanomagnet[49] ($\beta_{ac} = 2\lambda J_{ac}/J_p$, with $J_{ac}$, $\lambda$, $\boldsymbol{e}_p$ being injected current density, spin polarization factor, polarizer unit-vector, respectively) and $J_p = |e|\gamma M_s^2 t_{\text{FL}}/(g_L \mu_B)$ is an intrinsic current density value depending on free layer saturation magnetization $M_s$ and thickness $t_{\text{FL}}$ ($e$ is the electron charge, $g_L \approx 2$ is the Landé factor, $\mu_B$ is the Bohr magneton), $\nu$ is the thermal noise intensity, and $\boldsymbol{h}_{\text{N}}(t)$ is the standard isotropic Gaussian white-noise stochastic process. The energy function $g$, measured in units of $\mu_0 M_s^2 V$

($\mu_0$ is the vacuum permeability and $V$ the volume of the particle), is given by:

$$g(\boldsymbol{m}, \boldsymbol{h}_a) = \frac{1}{2} D_x m_x^2 + \frac{1}{2} D_y m_y^2 + \frac{1}{2} D_z m_z^2 - \boldsymbol{m} \cdot \boldsymbol{h}_a \quad (2)$$

where $D_x$, $D_y$, $D_z$ are the biaxial anisotropy constants. The intensity of the thermal noise $\nu$ is connected to the damping $\alpha$ in accordance with the fluctuation–dissipation theorem[49], i.e., $\nu^2 = (2\alpha k_B T)/(\mu_0 M_s^2 V)$, where $T$ is the absolute temperature of the thermal bath and $k_B$ is the Boltzmann constant.

We initially solve the Landau–Lifshitz equation in the absence of thermal noise ($\nu = 0$) in order to elucidate the role of deterministic chaos for the system of interest. Magnetization on the unit sphere described by Eq. (1) (with $\nu = 0$) is a two-dimensional dynamical system of nonautonomous type since the right-hand-side of the equation depends explicitly and periodically on time. This type of dynamics can be conveniently studied by introducing the stroboscopic map $P[\cdot]$, defined as[50]:

$$\boldsymbol{m}_{n+1} = P[\boldsymbol{m}_n], \quad (3)$$

where $\boldsymbol{m}_n = \boldsymbol{m}(t_0 + nT_{ac})$, and $T_{ac} = 2\pi/\omega$, which maps an initial magnetization $\boldsymbol{m}(t_0)$ to the magnetization $\boldsymbol{m}(t_0 + T_{ac})$ obtained by integrating Eq. (1) (with $\nu = 0$) over a time interval equal to the period $T_{ac}$ of the ac drive.

The mathematical form of the stroboscopic map cannot be derived in closed form, but certain features of the map dynamics can be obtained when the damping and the applied spin torque are small. In this case, the map describes the perturbation of the conservative dynamics described by Eq. (1) when $\alpha = 0$, $\beta_{ac} = 0$, and $\nu = 0$. The time evolution of magnetization in the conservative dynamics follows the constant energy contours, which are sketched in Fig. 3a. A crucial role in the conservative dynamics, Eq. (3) with $\alpha = \beta_{ac} = 0$, is played by the saddle equilibria points $(\boldsymbol{x}_{d1}^{(0)}, \boldsymbol{x}_{d2}^{(0)})$ heteroclinically connected by separatrices that mark the boundaries of the two potential wells (Fig. 3a).

For non-zero damping and spin torque ($\alpha \neq 0$, $\beta_{ac} \neq 0$), the saddle points of the map ($\boldsymbol{x}_{d1}$, $\boldsymbol{x}_{d2}$) are the origin of lines, referred to as stable and unstable manifolds, that play analogous role to the separatrices of the conservative case ($\alpha = 0$, $\beta_{ac} = 0$) and thereby provide structure to the state space (see Fig. 3b). The stable manifolds $W_1^s$, $W_2^s$ are sets (curves) of all initial conditions which under the action of the map (Eq. (3)) approach the saddles $\boldsymbol{x}_{d1}$, $\boldsymbol{x}_{d2}$, respectively. The unstable manifolds $W_1^u$, $W_2^u$ are sets (curves) of all initial conditions which under backward flow of time on the stroboscopic map (Eq. (3)) approach the saddles $\boldsymbol{x}_{d1}$, $\boldsymbol{x}_{d2}$, respectively. These manifolds are invariant sets, which means that they contain all forward and backward map iterates of points taken on them.

In Fig. 3b, the two manifolds $W_1^s$ and $W_2^u$ are sketched and their splitting is indicated by $d$. This splitting depends on the value of damping and ac spin torque, and it may vanish for a sufficiently large ac spin torque amplitude. When this occurs, a point of intersection $\boldsymbol{x}_a$ belonging to both invariant sets $W_1^s$ and $W_2^u$ emerges (see Fig. 3c). This implies that forward and backward iterates of $P[\cdot]$ starting from $\boldsymbol{x}_a$ must belong to $W_1^s \cap W_2^u$ and thus that the two curves $W_1^s$, $W_2^u$ must intersect an infinite number of times (see Fig. 3c). This phenomenon is referred to as heteroclinic tangle (chaotic saddle) and is responsible for chaotic and unpredictable dynamic behavior of the system near the saddles. This chaotic dynamics can be illustrated in terms of lobe dynamics. Regions of the state space bounded by segments of stable and unstable manifolds of the saddles form lobes, examples of which are the colored regions in Fig. 3c. Under the action of the map, one lobe transforms into another. There are two classes

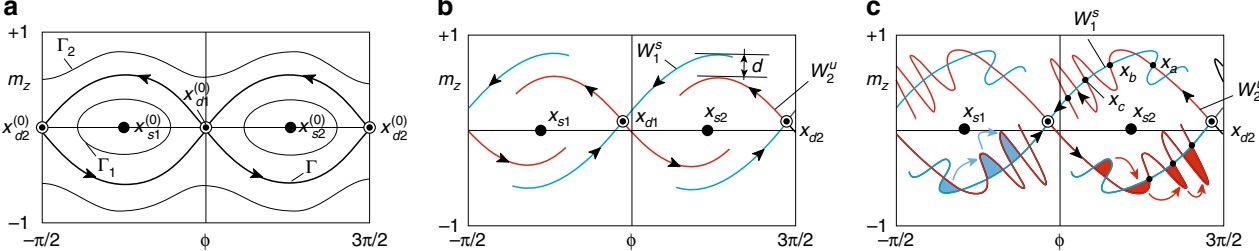

**Fig. 3** Qualitative sketches of the magnetization trajectories in the ($\phi$, $m_z$)-plane. ($\phi$ is the azimuthal angle around the $z$-axis). **a** Conservative trajectories ($\alpha = 0$, $\beta_{ac} = 0$, designated by superscript (0)). $\Gamma_1$, $\Gamma_2$ are constant energy trajectories and $\Gamma$ is the heteroclinic trajectory (separatrix). **b** Damping dominated dynamics ($\alpha > 0$, $\beta_{ac} < \beta_x^{crit}$, $d > 0$). Here $x_{d1}$, $x_{d2}$ are saddle equilibria; $x_{s1}$, $x_{s2}$ are node-type equilibria; $W_1^s$ is stable manifold associated with $x_{d1}$; $W_2^u$ is unstable manifold associated with $x_{d2}$; $d$ is the splitting of the manifolds. **c** Heteroclinic tangle formation ($\alpha > 0$, $\beta_{ac} > \beta_x^{crit}$). The manifold intersection points $x_a$, $x_b$, $x_c$ are generated by iterating the stroboscopic map, Eq. (3). Intersecting stable and unstable manifolds form lobes. Blue regions indicate capturing lobes that keep the magnetization inside the given potential well while red regions show escaping lobes that bring magnetization outside of the well. The colored arrows indicate the transformation of one lobe into another under the action of the map, Eq. (3)

of lobes: the escaping ones (marked by red color), which tend to bring points outside the well, and the capturing lobes (marked by blue color), which tend to bring points inside the well. Escaping and capturing lobes do actually finely intersect possibly a denumerable amount of times, and this gives rise to a fractal boundary between the points which enter the well and points which escape the well[50]. The region in which lobes formed by the stable and unstable manifold intersect is the region where chaotic saddle dynamics takes place.

For sufficiently small applied torques and damping, the splitting $d$ can be analytically derived by using the Melnikov function technique[51] and one can calculate the threshold ac torque[52] at which the splitting becomes zero and the saddle becomes chaotic. Such threshold values of $\beta_{ac}$ for the onset of the heteroclinic tangle, for spin current polarized along each of the anisotropy principle axes are[41]:

$$\beta_x^{crit} = \beta_{opt}^{crit}/k', \quad \beta_y^{crit} = \infty, \quad \beta_z^{crit} = \beta_{opt}^{crit}/k,$$
$$\beta_{opt}^{crit} = \frac{2\alpha\Omega_d}{\pi}\cosh\frac{\pi\omega}{2\Omega_d}, \tag{4}$$

where $k^2 = (D_z - D_y)/(D_z - D_x)$, $k'^2 = 1 - k^2$, $\Omega_d = \sqrt{(D_z - D_y)(D_y - D_x)}$. The infinite result for the case of $y$-polarization is due to the fact that the method is first order accurate with respect to perturbation amplitudes $\alpha$, $\beta$. The result implies that a spin polarization along the $y$-axis produces a much weaker effect with respect to other orientations. In our experiment, the spin polarization vector set by the SAF magnetic moment direction is parallel to the $x$-axis, which means that the critical ac spin torque value needed to induce chaotic magnetization dynamics in our MTJ system is $\beta_x^{crit}$.

Using Eq. (4), we calculate the zero-temperature threshold ac voltage for the onset of heteroclinic tangle $V_{ac}^0(f)$. For the conversion from dimensionless to physical units, we remark that a spin–torque amplitude $\beta_x^{crit} = 1$ corresponds to an ac voltage $V_{ac}^0 = 15.5$ V and a dimensionless angular frequency $\omega = 1$ corresponds to a frequency $f = 30.8$ GHz (see Methods for details). Thus, the calculated threshold voltage $V_{ac}^0(f)$ is compared to the measured threshold voltage in Fig. 2c.

## Discussion
The amplitude and frequency dependence of measured threshold voltages are consistent with those theoretically predicted in the 1–4 GHz frequency range (region labeled Deterministic chaos in Fig. 2c). The deviations of the threshold voltage from the value predicted by the heteroclinic tangle theory at frequencies below 1

GHz (region labeled Néel–Brown in Fig. 2c) arise from adiabatic enhancement of the amplitude of thermal fluctuations of magnetization by spin torque. When the ac spin torque frequency is lower than the Néel–Brown attempt frequency for thermally activated switching[53], antidamping spin torque can significantly amplify the amplitude of thermally-induced magnetization precession in the half-cycle of the ac drive when spin torque acts as antidamping[54] and thereby induce switching over the energy barrier. It has been previously shown that efficient amplification of the free layer magnetization amplitude by ac spin torque drive is limited to frequencies below the free layer magnetization relaxation rate, which typically does not exceed 1 GHz[55].

This mechanism of thermally assisted switching is distinctly different from the heteroclinic tangle mechanism and it can significantly decrease the low-frequency value of the ac threshold voltage below that predicted by Eq. (4). To verify the role of this mechanism, we numerically solved[56] the stochastic LL equation (Eq. (1)) at $T = 300$ K (Methods). The results of these simulations shown in Fig. 4b are in a remarkably good agreement with the experimental data shown in Fig. 4a over the entire frequency range employed in the experiment. These simulations confirm that the threshold voltage is significantly reduced by the antidamping action of ac spin torque at frequencies below the attempt frequency but is nearly insensitive to temperature at frequencies exceeding the attempt frequency. We also note that ac drive increases the sample temperature via Joule heating. However, Joule heating is frequency independent and thus cannot explain the observed frequency dependence of the switching rate.

We also observe deviations of the threshold voltage from the value predicted by the heteroclinic tangle theory at frequencies near the FMR frequency (region labeled FMR in Fig. 2c), which can be explained by resonant transfer of energy from the ac spin torque to magnetization at the FMR frequency[14]. This resonant mechanism of lowering the external field required to reach the thermally assisted switching regime of the otherwise stable magnetization is exploited in MAMR technologies[30–32]. It has been shown that MAMR can be optimized for more efficient magnetization reversal when comparing the consumed energy of dc field switching alone to dc + ac switching[31]. From the data in Fig. 4, one can see that the erosion of the effective energy barrier due to chaos at sub-FMR frequencies is more efficient at leading to magnetization switching than the resonant absorption of energy near the FMR frequency. This implies that deterministic chaos induced by a sub-FMR-frequency drive may be a more energy-efficient approach to lowering the external field required for energy-assisted switching of magnetization than MAMR at the FMR frequency. Such an approach may additionally

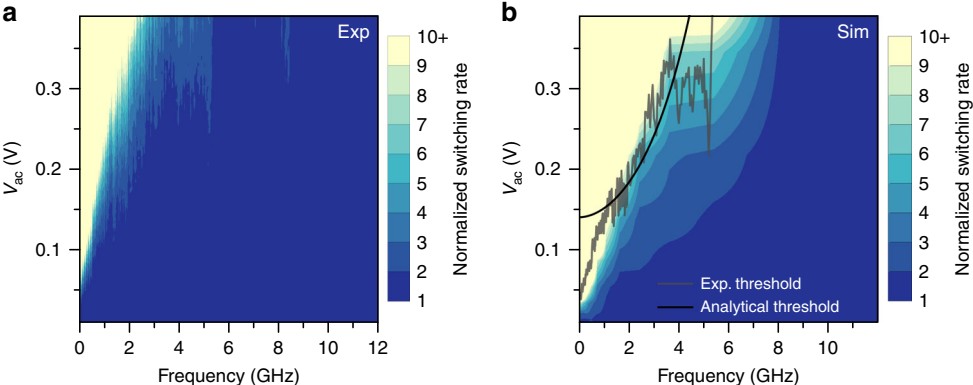

**Fig. 4** Room temperature free layer switching rate. The rate is mapped as a function of the applied ac voltage $V_{ac}$ and frequency $f$ for **a** experiment and **b** numerical solution of the stochastic LL equation (Eq. (1)). The black line overlaid on (**b**) is analytical prediction of the threshold voltage for the onset of zero-temperature chaotic dynamics due to heteroclinic tangle, while the gray line is experimentally measured threshold voltage obtained from the experimental data in (**a**)

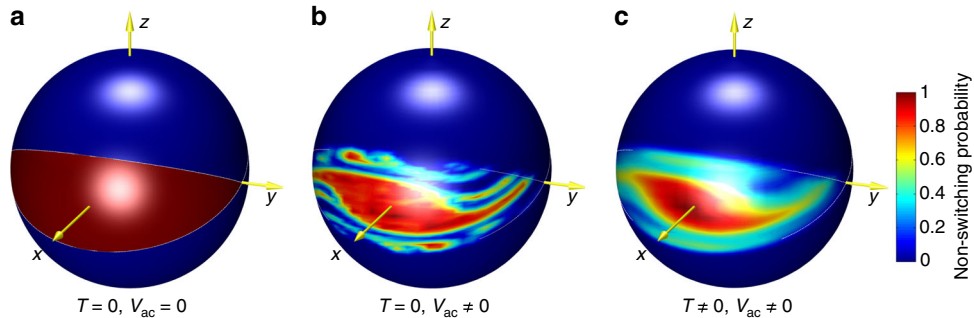

**Fig. 5** Visualizing chaos. Directional map of probability of magnetization initially within the $x > 0$ potential well to remain within this well after 5 periods of alternating spin torque drive. Red color marks the initial direction of magnetization resulting in magnetization staying within the well, while blue color marks initial magnetization direction resulting in escape from the well after 5 periods of the drive. **a** At zero temperature ($T = 0$) and in the absence of spin torque, any initial direction of magnetization within the well indefinitely remains within the well. **b** At zero temperature and alternating spin torque exceeding the threshold value, the fractal structure of escaping and capturing lobes within the well near the separatrices is apparent. Colors intermediate between red and blue appear as a result of coarse grained averaging of the fractal lobe structure over a small but finite solid angle. It is clear from this figure that chaotic dynamics induced by alternating spin torque decreases the basin of stability to a region in the center of the well, thereby reducing the effective energy barrier between the two wells. **c** At room temperature and spin torque drive exceeding the threshold value, the fractal lobe structure of the well is blurred by thermal fluctuations but the chaos-induced erosion of the basin of stability is still apparent

decrease the dc current needed and power efficiency for switching of STT-RAM.

It is important to stress that the effect observed in the 1–4 GHz regime is not related to the FMR phenomenon. In FMR type dynamics, magnetization exhibits ac-driven small-angle deviations from the easy axis $m_x = \pm 1$, and the ac frequency is close to the FMR frequency. In the present study, we focus on dynamics driven by frequencies substantially lower than the FMR frequency. For sufficiently high ac power, chaotic dynamics is induced when magnetization is almost orthogonal to the easy axis (near $m_y = \pm 1$, the saddle critical points of anisotropy energy, as illustrated in Fig. 1b). This regime corresponds to the top of the two potential wells in Fig. 1c, which are rarely visited by magnetization in thermal equilibrium. However, in the process of switching from one well to the other, magnetization must pass near one of the saddle critical points and, for this reason, the switching rate is affected by the ac-driven chaotic dynamics.

From another perspective, such chaotic regime results in partial reduction of the magnetic anisotropy barrier separating two potential wells[57]. The definition of an effective potential in ac-driven regime is not straightforward and was only derived in the limit of weak noise[58] that is not applicable to our superparamagnetic MJT system. Nevertheless, the reduction of the

stability margin of the equilibrium can be quantified in terms of erosion. The concept of such barrier erosion, as well as a more detailed picture of the interplay between chaotic dynamics and thermal fluctuations, are illustrated in Fig. 5a–c, which are obtained by numerical simulations of magnetization dynamics (Supplementary Note 3 and Supplementary Figure 4). These figures can be thought of as the actual representation of the qualitative sketch shown in Fig. 1c. By using a color scale, we represent the different degrees of stability of the magnetization states inside a potential well. The degree of stability of a given state is measured by the number of ac-excitation periods after which the trajectory starting from that state leaves the $m_x = +1$ potential well. In this figure, red points represent the initial directions of magnetization that do not leave the potential well and, therefore, are states with the highest stability; conversely, points with color toward the blue are those with decreasingly less stability.

In Fig. 5a, for zero temperature and zero ac excitation, stable points fill the entire energy well around the energy minimum along the easy $x$-axis. In Fig. 5b, for zero temperature and sufficiently large ac voltage, fractal instability regions arising from chaotic dynamics appear in the energy well, which reduces the stability margin of the equilibrium. In Fig. 5c, when both ac voltage and temperature are nonzero, the entangled instability

regions due to chaos are smoothed out by thermal fluctuations, but the effective erosion of stability margin of the equilibrium remains, which implies a reduction of the barrier for thermally activated switching. With increasing degree of erosion, the thermally activated switching rate dramatically increases. This is the experimentally detectable signature of chaos.

The interplay between such chaotic regime of magnetic dynamics and thermal fluctuations, to our knowledge, has not been studied. It is well-known that the thermal transition (escape) times are described by the Arrhenius law: $\tau = \tau_0 \exp[\Delta E/k_B T]$, where $\Delta E$ is the anisotropy energy barrier for switching, and $1/\tau_0$ is the attempt frequency. The problem of generalizing the Arrhenius law to the regime of ac-driven chaotic dynamics remains largely open. Our work shows that the effect of chaotic dynamics on the escape rate is detectable in magnetic nanodevices, which should stimulate both experimental and theoretical studies of this important problem.

In contrast to the majority of previous studies of chaotic dynamics focused on steady-state chaotic motion, the chaotic dynamics studied here is of transient type[59]. This kind of chaos is essentially of the same nature as the one discovered by Poincaré in the three-body problem and addressed by KAM theory. However, the magnetic chaos studied here cannot be directly described by KAM theory, which is formulated for conservative systems. Indeed, ac spin torque driving chaotic magnetic dynamics in our MTJ system is manifestly non-conservative (as well as the Gilbert damping torque).

Transient chaos plays important role in a wide range of phenomena. It is key to the understanding of planetary system stability and for describing the accretion, escape, and scattering of planetesimals forming planetary embryos[7]. The complex behavior of ecological systems such as food chain dynamics and recurrence of epidemic diseases often requires a description in terms of transient chaos[5]. Certain optomechanical systems exhibit apparent breakdown of the quantum-classical correspondence[60], wherein the classical regime dynamics of the system is chaotic while the quantum dynamics is regular. Recently, it was shown that transient chaos could resolve this paradox and explain the quantum-classical correspondence in these systems[61]. Our results advance the understanding of transient chaos by using a low dimensional and controllable real-world system.

Recent theoretical work has shown that interesting magnetization dynamics can also manifest in a uniformly magnetized domain under large ac external fields in ways that are not chaotic[19,62]. It has been shown that large ac drive can lead to large enhancements of transient behavior times as well as a different kind of magnetization reversal where a dynamic state with magnetization opposite to the applied dc field is achieved. There are features of this type of dynamics that are very distinct from the chaotic phenomena reported in this article. First, the phenomena we observe has a monotonic threshold behavior as a function of drive amplitude, while the effect reported in refs. [19,62] is non-monotonic. Second, the magnetization reversal achieved by our mechanism on the bistable magnetization is persistent upon removing ac drive (up to thermal activation time scales), while the dynamic reversed state achieved by the other mechanism is not. Finally, the two mechanisms have distinctly different dependence on ac drive frequency.

As a general rule, physical systems with multistable energy landscape, which are weakly dissipative and subject to sufficiently large ac excitations, exhibit chaotic dynamics near their saddle equilibria[50]. Nanoscale magnetic devices, such as the MTJs studied in this work, are an important case of this class of systems. The possibility of measuring and controlling magnetization in these devices in real time gives a unique opportunity to study chaotic dynamics. In this work, we explored this opportunity

which, to our knowledge, is one of the first clear attempts to detect chaos in nanoscale systems at room temperature. It is remarkable that the analytical calculation based on deterministic dynamics leads to correct predictions of the experimentally measured frequency dependence of the threshold power needed to trigger low-dimensional magnetic chaos. Our results are not restricted to the field of nanomagnetism; we expect them to be important in a variety of driven dynamical systems and phenomena such as nanomechanical oscillators[63], multistable lasers[64], and stimulated chemical reactions[65].

Finally, from the application point of view, energy-efficient switching of magnetization is highly desirable for practical spintronic memory and logic devices[66,67]. Our work shows that ac-driven chaos can facilitate thermally-assisted switching of magnetization, which provides a pathway towards energy-efficient magnetic nanodevices. Nanoscale MTJs with superparamagnetic free layers are now being exploited in emergent neuromorphic and reservoir computing[33,43,68]. While we have demonstrated the effect in nanoscale MTJs with superparamagnetic free layers, ac-driven chaos is also expected to facilitate switching of thermally-stable free layers employed in non-volatile memory applications. Our results show that low-dimensional chaos provides tunability of switching rates in such systems, and as such, may lead to computing schemes that simultaneously harness stochasticity and deterministic chaos.

## Methods

**Sample description**. The MTJs are patterned from (bottom lead)|(5)Ta|(15)PtMn| SAF|(0.8)MgO|(1.8)CoFeB|(2)Cu| (top lead) multilayers (thickness in nm) deposited by magnetron sputtering. Here SAF ≡ (2.5)$Co_{70}Fe_{30}$|(0.85)Ru|(2.4)$Co_{40}Fe_{40}B_{20}$ is the pinned SAF, with magnetic moments lying in the sample plane, which is used as both the polarizer and reference layer. The unpatterned multilayers were annealed at a temperature of 300 °C in an external field of 1 T applied in the sample plane for 2 h. The elliptical MTJs are patterned such that the major axis (easy ferromagnetic axis) is parallel to the annealing field direction.

**Spin torque ferromagnetic resonance**. We employ field-modulated ST-FMR technique to determine $f_{FMR}$ in our samples at room temperature ($T = 300$ K)[45,69,70]. In these measurements, microwave voltage is applied to the MTJ through the ac port of the bias tee, and rectified voltage generated by the MTJ at the frequency of magnetic field modulation is measured by a lock-in amplifier through the DC port of the bias tee, as schematically illustrated in Fig. 2a.

**Random telegraph noise measurements**. The thermally activated switching rate of the free layer in the MTJ is monitored via RTN measurements at room temperature ($T = 300$ K). The experimental setup for the RTN measurements is shown in Fig. 2a. A low-level probe current ($-25$ μA) is applied to the MTJ and the voltage across the device is measured by a high-performance DAQ (National Instruments USB-6356) in order to monitor the MTJ resistance as a function of time. In these measurements, we apply a small in-plane magnetic field (3.7 mT) along the nanomagnet easy axis that compensates the stray field from the SAF layer acting onto the free layer and balances the dwell times of the free layer in the high-resistance (AP, 3350 Ω) and low-resistance (P, 1450 Ω) states. A microwave frequency ac voltage can be applied to the MTJ via the ac port of the bias tee. This voltage gives rise to an ac spin torque applied to the free layer by spin-polarized electric current from the SAF layer. The dwell times of the P state $\tau_P$ and the AP state $\tau_{AP}$ were found to remain balanced under the ac drive ($\tau_{AP} = \tau_P = \tau$). The switching rate of the free layer nanomagnet is the inverse of the dwell time $w \equiv 1/\tau$.

**Numerical simulations**. The stochastic LL equation (Eq. (1)) has been repeatedly solved[56] for an ensemble of $N = 20{,}000$ particle replicas, for given values of $\beta_{ac}$ and $\omega$ to compute the average switching rates of Fig. 4b. The values of parameters used in simulations are $\mu_0 M_s = 1.1$ T, $D_x = 0.035$, $D_y = 0.056$, $D_z = 0.909$, $\alpha = 0.016$. Dimensionless angular frequency $\omega = 1$ corresponds to frequency $f = \gamma M_s/(2\pi) = 30.789$ GHz, dimensionless ac spin–torque $\beta_{ac} = 1$ corresponds to injected ac current with amplitude $I_{ac} = J_P S/(2\lambda) = 6.45$ mA (polarization factor $\lambda = 0.6$, MTJ cross-sectional area $S = 2.9452 \times 10^{-15}$ m$^2$, $J_P = |e|\gamma M_s^2 t_{FL}/(g_L \mu_B) = 2.63 \times 10^{12}$ Am$^{-2}$, $t_{FL} = 1.8$ nm). Conversion between current and voltage applied to the MTJ is performed as $V_{ac} = RI_{ac}$ where the resistance $R = (R_P + R_{AP})/2 = 2400$ Ω is the average between the measured MTJ resistance values in the parallel and anti-parallel states; thus, an ac spin–torque $\beta_{ac} = 1$ corresponds to a voltage $V_{ac} = 2400$ Ω $\times$ 6.45 mA = 15.5 V.

**Code availability**. Micromagnetic custom code can be made available from M. d'Aquino and C. Serpico upon reasonable request.

## Data availability

All data supporting the findings of this study are available within the article and the Supplementary Information file, or are available from the corresponding author on reasonable request.

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

## Acknowledgements

This work was supported by the National Science Foundation through Grant Nos. DMR-1610146, EFMA-1641989, and ECCS-1708885. The authors also acknowledge support by the Army Research Office through Grant No. W911NF-16-1-0472 and Defense Threat Reduction Agency through Grant No. HDTRA1-16-1-0025. This work was also carried on in the framework of Programme for the Support of Individual Research 2017 funded by the University of Naples Parthenope. The authors thank Juergen Langer and Berthold Ocker for magnetic multilayer deposition.

## Author contributions

I.N.K., M.d'A., and C.S. planned the study. E.A.M. designed and performed RTN measurements and performed ST-FMR measurements with consultation from Y.-J.C. M.d'A. and C.S. provided numerical simulation code and analytic theory. S.P. performed simulations under the supervision of M.d'A. J.A.K. made the samples. M.d'A., I.N.K., C.S., and E.A.M. wrote the manuscript. All authors discussed the results.

## Additional information

**Competing interests:** The authors declare no competing interests.

