## [Peer Review File · Nature Communications]

Reviewers' comments:

Reviewer #1 (Remarks to the Author):

This is a very nice paper focusing on the thermal switching behavior of a magnetic nanoelement when driven by a ac spin current.

The authors point out, correctly, that there has not been much research in this area, particularly in regards to the interplay between chaotic and thermal behaviors. The combination between theory and experiment is convincing and enlightening.

There are some things, however, that the authors could address:

1) Perhaps I missed it, but I didn't see any reference to the temperature at which the experiment is carried out. The simulations are a 300 K, so I assume the experiment is as well. But this does raise a couple of questions: (a) is there some heating from the current? (b) can the experiment be done at a different temperature so as to improve the understanding of the temperature-chaos interplay?

2) The authors point out that their results show that the ac spin current is in a sense more efficient than MAMR. While this is true in this specific case, it seems a bit misleading. In most cases one thinks of MAMR as a potential way to improve on magnetic storage. But for storage, one does not want to be near a temperature range where there is thermal switching. It seems a sentence or two in cautioning the reader would be appropriate.

3) If one is just interested in obtaining the lowest threshold voltage – the data in this paper argues that one should just use a very low frequency current, i.e. just a short dc current pulse would be fine. It wasn't clear that there is actually any practical advantage of the ac current versus dc current.

I emphasize that the physics in the ac case is interesting, so what is needed is a clearer picture of why ac chaotic reversals are better than low frequency reversals in an application.

4) The authors state that "The process of magnetization switching from one potential well into the other necessarily involves crossing the separatrices and, therefore, must proceed via the band of chaotic dynamics induced by the ac drive. Therefore, we expect the ac-driven chaotic dynamics to affect the nanomagnet switching behavior." There are, however, a couple of recent papers that point out a kind of magnetization reversal is possible without chaos. (Phelps et. al. in EPL Vol 9, 37007 (2015) and Feron Phys Rev B 95, 104421 (2017)) The authors might want to address the differences.

Reviewer #2 (Remarks to the Author):

The authors presented a very interesting work about using alternating spin transfer torque (STT)-driven, low-dimensional magnetic chaos to increase the rate of thermally-activated magnetization switching in a nanoscale ferromagnetic element. The experimental data are new to the best of my knowledge, and are supported by analytical and numerical analyses. This work should be of great interest to the spintronics and magnetic memory communities.

I have the following questions and suggestions for the authors:

In Fig. 2c, the theoretical curve clearly shows a positive curvature while the experimental data do not. Can the authors explain this difference? On page 5, the authors said that the agreement between the theoretical and experimental data on Fig. 2c is "excellent." I would suggest the

authors revise this statement.

Regarding the roles of the alternating spin torque-produced chaos in assisting the magnetization switching, the authors stated that the chaos reduces the stability margin of the equilibrium. This is shown in Fig. 5. However, the authors also mentioned that the chaos leads to a reduction of the energy barrier for thermally activated switching, on page 6. Does the chaos result in both “a reduction of the stability margin of the equilibrium” and “a decrease in the energy barrier for the thermally activated switching”? Or, just one of them?

The authors call Eq. (1) the Landau-Lifshitz equation on page 4 but call it the LLG equation on pages 5, 6, and 10. I wish the authors could be consistent, as the LL equation differs from the LLG equation in several different ways. I would think the equation is the LL equation, not the LLG equation.

The authors emphasized several times that the chaos involved in the switching is low-dimensional. How low? Can the authors estimate the fractal dimension of the chaos?

The authors mentioned several times that the CoFeB thin film is superparamagnetic, but for me the film is pretty much the same as the free layers in typical MTJ devices. Why is it superparamagnetic? Did the authors carry out field-cooled and zero-field-cooled measurements to determine the blocking temperature?

On page 2, the authors wrote “microwave-assisted magnetic recording (MAMR) relies on thermally activated switching of nanoscale ferromagnets.” This may not be an accurate statement, because MAMR mainly utilizes an external microwave to reduce the energy barrier and an external static field to realize deterministic switching.

On page 6, the authors wrote “deterministic chaos induced by a sub-FMR-frequency drive may be a more energy-efficient approach to energy-assisted switching of magnetization than MAMR at the FMR frequency.” I am wondering how the frequency of the magnetization precession during the switching is compared to the microwave frequency? In case they are close, the film may also take energy from the microwave, as in MAMR.

Does microwave heating play any roles in the presented switching experiments, especially in the regime where high-power microwaves were used (Fig. 2c)?

In Fig. 5, the parameter T is not defined. Change “withing” to “within”.

Reviewer #3 (Remarks to the Author):

The results presented in the manuscript entitled “Magnetization reversal driven by low dimensional chaos in a nanoscale ferromagnet” might, indeed, be essential for the development of the emerging spintronic-based stochastic and reservoir computing devices. Furthermore, if communicated in an accessible way, the interpretation of the observe stochastic phenomena should appeal to the experts from other research fields and general audience. Unfortunately, the way the manuscript is written now does not easily allow to grasp the essence of the observed effect, in particular for the non-expert in the chaotic dynamics. I believe that the interpretation based on the stochastic resonance phenomena (which some of the co-authors are experts in) would be less complicated, and, thus, more appealing to prospective readers of Nature Nanotechnology. Stochastic resonance has been observed and understood in spintronic devices before, including the papers co-authored by the authors of the present manuscript that raises a question about the novelty of the results. So in the revised manuscript, it should be clearly stated how present results advance understanding of stochastic resonance in spintronic devices. The second major remark is regarding the theory developed in the manuscript. It appears to me (Fig.

2c) that experimentally measured V_{ac}^0 vs. frequency (f) shows some \sqrt{f} behavior in the entire range of frequencies, while theory predicts f^p , where $p > 1$. So, most likely, the macrospin theory cannot grasp the underlying stochastic process. Therefore I would like authors to discuss this discrepancy and, at least, mention its possible causes. Finally, there are at least three relatively slow processes that might explain enhanced switching rates at low injection frequencies:

- time needed for the phase oscillator to return on the limiting cycle after perturbation $1/\Gamma_p$, where Γ_p is amplitude-to-frequency conversion rate, which is typically below 1 GHz.
- (super)paramagnetic resonance frequency, $\omega = \gamma B = 0.11$ GHz for the parameters mentioned in the manuscript.
- Kramers rate that according to Fig.2a is around 100 Hz.

I would like authors to discuss possible contributions of these effects to the observed results.

Overall, although I believe the manuscript might be suitable for the publication in Nature Communications, I cannot recommend it for publication in its present form.

We would like to thank all the Reviewers for their time in evaluating and commenting on our manuscript. We believe that by following their suggestions the manuscript has been greatly improved. Please find below our point-by-point replies to the Reviewer comments. Our replies (indented blue text) are interlaced with the Reviewers' original comments. Additionally, we have also highlighted changes in the main manuscript document in blue text for easy identification.

Reviewer #1 (Remarks to the Author):

This is a very nice paper focusing on the thermal switching behavior of a magnetic nanoelement when driven by a ac spin current.

The authors point out, correctly, that there has not been much research in this area, particularly in regards to the interplay between chaotic and thermal behaviors. The combination between theory and experiment is convincing and enlightening.

We thank the Reviewer for positive evaluation of our work and for recognizing the importance of our theoretical and experimental results on the interplay between chaotic and thermal behaviors in magnetic nanoelements.

There are some things, however, that the authors could address:

1) Perhaps I missed it, but I didn't see any reference to the temperature at which the experiment is carried out. The simulations are a 300 K, so I assume the experiment is as well. But this does raise a couple of questions: (a) is there some heating from the current? (b) can the experiment be done at a different temperature so as to improve the understanding of the temperature-chaos interplay?

All measurements are performed at room temperature. In the revised manuscript, we made sure to state the measurement temperature in both the main text and the Methods section.

(a) There is ohmic heating induced by the current. However, the heating at different microwave frequencies is the same for a given microwave power (ensured by calibration of our microwave circuit), and thus the measured frequency dependence of the switching rate cannot be explained by ohmic heating.

(b) In principle the experiment can be performed at a different temperature. However the temperature range accessible by our technique based on random telegraph noise is very limited and thus the measurements would not be very revealing. For lower temperatures, the Kramers transition rate becomes exponentially slow leading to prohibitively long data collection times. Upon heating, the switching rate becomes exponentially fast and exceeds the bandwidth of the experimental setup (25 MHz).

2) The authors point out that their results show that the ac spin current is in a sense more efficient than MAMR. While this is true in this specific case, it seems a bit misleading. In most cases one thinks of MAMR as a potential way to improve on magnetic storage. But for storage, one does not want to be near a temperature range where there is thermal switching. It seems a sentence or two in cautioning the reader would be appropriate.

We agree with the Reviewer that magnetic particles in MAMR hard drives are thermally stable and thus are far from the superparamagnetic regime employed in our studies. Nevertheless the magnetic reversal in MAMR is, to a lesser degree, thermally assisted (as is always the case for a bistable

nanomagnet at a non-zero temperature) and thus the effective barrier for thermally activated switching measured by our technique is relevant to MAMR. Our choice of the superparamagnetic regime for measurements of this barrier is merely a matter of convenience allowing us to observe many switching events within a reasonably short measurement time. We agree with the Reviewer that we did not discuss the degree of relevance of our measurements to MAMR in sufficient detail, and thus we added such a discussion on page 6 of the revised manuscript. We changed the language to more clearly point out that MAMR is used in conjunction with an external field to lower the strength of the external field required for sufficient probability of switching. Since the chaotic dynamics induced by the ac drive also lead to an effective reduction of the energy barrier, this implies that chaotic mechanism also leads to a reduction of the external field needed for switching.

3) If one is just interested in obtaining the lowest threshold voltage – the data in this paper argues that one should just use a very low frequency current, i.e. just a short dc current pulse would be fine. It wasn't clear that there is actually any practical advantage of the ac current versus dc current.

I emphasize that the physics in the ac case is interesting, so what is needed is a clearer picture of why ac chaotic reversals are better than low frequency reversals in an application.

This is a very good question that warrants further discussion and studies. The main physics result of our work is demonstration that ac-driven chaos can induce switching of magnetization. That said, such a novel effect may find use in magnetic nanodevices for technological applications, which requires further detailed studies. Here we outline the reason why ac drive may be beneficial for switching in spite of the minimum of threshold voltage found in the limit of zero drive frequency.

To achieve energy-efficient switching, one must minimize the total power consumed in the switching process. Given that the dominant losses are usually ohmic, such power is proportional to the applied voltage squared multiplied by the applied voltage pulse duration. Since higher voltage leads to faster switching, there is a tradeoff between higher voltage and shorter write pulse duration. In the case of deterministic spin torque switching of MTJs, the minimum switching power is achieved when the applied voltage pulse amplitude is twice the critical voltage ($V=2*V_0$) (Zhao, et. al. J. Appl. Phys. 109, 07C720 (2011), substitute Eq. 2 into Eq. 4 and minimize). This illustrates that one may need to go significantly above the critical voltage to achieve the optimal energy efficiency.

It has been shown that the external field required for switching using MAMR is greatly reduced (Zhu et al. IEEE Trans. Magn. 44 125-131 (2008)). This can lead to more efficient magnetization reversal. For example, Zhu et al. show that for a reversal field at 30 degrees from easy axis and an ac field equal to 0.1 Hk, the reversal field is reduced from 0.525 Hk to 0.15 Hk. The total energy required for reversal is proportional to the square of the dc reversal field plus the square of the RMS amplitude of the ac field. In this situation, the total energy in sum of fields is reduced by a factor of 10. A similar effect may be present in the case of spin torque switching of MTJs: application of a microwave pulse, which amplitude (but not necessarily its RMS amplitude) exceeds the critical voltage for chaos-induced barrier reduction, combined with a dc pulse with $V < 2*V_0$ may be more energy efficient than a single dc voltage pulse of $2*V_0$ amplitude. Unfortunately, testing of this hypothesis requires extensive numerical calculations because of the lack of analytical expression for chaos-induced effective barrier reduction. This practically important but complicated problem goes beyond the scope of present work. In the revised manuscript, we highlight the possibility of increased spin torque switching efficiency in a manner similar to MAMR to inspire future theoretical and experimental work (revision on page 6).

4) The authors state that “The process of magnetization switching from one potential well into the other necessarily involves crossing the separatrices and, therefore, must proceed via the band of chaotic dynamics induced by the ac drive. Therefore, we expect the ac-driven chaotic dynamics to affect the nanomagnet switching behavior.” There are, however, a couple of recent papers that point out a kind of magnetization

reversal is possible without chaos. (Phelps et. al. in EPL Vol 109, 37007 (2015) and Feron Phys Rev B 95, 104421 (2017)) The authors might want to address the differences.

We thank the Reviewer for pointing out this inaccuracy. Indeed, with this sentence we did not mean to claim that ac-driven chaos is the only possible mechanism of switching. To correct the problem, we modified these sentences to: "The process of magnetization switching from one potential well into the other proceeds via the band of chaotic dynamics induced by the ac drive, when such a chaotic band is generated at sufficiently large ac drive amplitude. Therefore, we expect the ac-driven chaotic dynamics excited above appropriate threshold amplitude to affect the nanomagnet switching behavior" Additionally in the revised manuscript, we now discuss other types of ac-driven magnetization reversal that do not involve chaos, in particular those described in (Phelps et. al. in EPL Vol 109, 37007 (2015) and Feron Phys Rev B 95, 104421 (2017)). This discussion is given on page 8 of the revised manuscript.

Reviewer #2 (Remarks to the Author):

The authors presented a very interesting work about using alternating spin transfer torque (STT)-driven, low-dimensional magnetic chaos to increase the rate of thermally-activated magnetization switching in a nanoscale ferromagnetic element. The experimental data are new to the best of my knowledge, and are supported by analytical and numerical analyses. This work should be of great interest to the spintronics and magnetic memory communities.

We thank the Reviewer for their time and positive evaluation of our work and for recognizing its potential interest to the spintronics and magnetic memory communities.

I have the following questions and suggestions for the authors:

In Fig. 2c, the theoretical curve clearly shows a positive curvature while the experimental data do not. Can the authors explain this difference? On page 5, the authors said that the agreement between the theoretical and experimental data on Fig. 2c is "excellent." I would suggest the authors revise this statement.

The reason positive curvature is not seen in the experimental data over the entire frequency range is that switching in the low-frequency regime (< 1 GHz) and the FMR regime (near 5 GHz) is dominated by mechanisms other than chaos-induced effective barrier reduction. In the frequency range where deterministic chaos is the dominant mechanism influencing the switching rate (1 GHz - 4 GHz) the agreement with the analytical curve derived from the deterministic chaos theory is good, and we numerically verified the curvature of the experimental data is positive in this frequency range. We have clarified this point in the revised manuscript, which now states on page 5, "The amplitude and frequency dependence of measured threshold voltages are consistent with those theoretically predicted in the 1-4 GHz frequency range (region labeled Deterministic chaos in Fig. 2c)."

Regarding the roles of the alternating spin torque-produced chaos in assisting the magnetization switching, the authors stated that the chaos reduces the stability margin of the equilibrium. This is shown in Fig. 5. However, the authors also mentioned that the chaos leads to a reduction of the energy barrier for thermally activated switching, on page 6. Does the chaos result in both "a reduction of the stability margin of the equilibrium" and "a decrease in the energy barrier for the thermally activated switching"? Or, just one of them?

Indeed, the expressions 'reduction of the stability margin' and 'decrease in the energy barrier' refer to two aspects of the same phenomenon - appearance of the band of chaotic dynamics above the threshold ac drive. On one hand, the reduction of the stability margin can be quantified by using the

concept of basin erosion as suggested by Thompson (Ref. 56), which can be computed numerically as we did in order to produce the maps in Fig. 5. On the other hand, the concept of energy barrier under ac drive requires a definition of an effective energy landscape in the presence of ac excitation. The general discussion of this issue is an active area of research and has not yet been fully settled theoretically. In fact, most results (e.g. R.L. Kautz et al., Physics Letters A 125, 315 (1987); R. Graham et al., Phys. Rev. Lett 66, 3089 (1991)) are derived under the assumption of weak noise, which is not the case in our paper. For these reasons, a precise quantitative relation between the basin erosion and effective energy barrier reduction in ac-driven regime is still lacking. In order to clarify this aspect in the paper, we expanded the discussion on pages 6-7 to become: "From another perspective, such chaotic regime results in partial reduction of the magnetic anisotropy barrier separating two potential wells [Thompson1989]. The definition of an effective potential in ac-driven regime is not straightforward and was only derived in the limit of weak noise [Kautz1987, Graham1991] that is not applicable to our superparamagnetic MTJ system. Nevertheless, the reduction of the stability margin of the equilibrium can be quantified in terms of erosion. The concept of such barrier erosion, as well as a more detailed picture of the interplay between chaotic dynamics and thermal fluctuations, are illustrated in Fig.5a-c, which are obtained by numerical simulations of magnetization dynamics (Supplementary Note 2)."

The authors call Eq. (1) the Landau-Lifshitz equation on page 4 but call it the LLG equation on pages 5, 6, and 10. I wish the authors could be consistent, as the LL equation differs from the LLG equation in several different ways. I would think the equation is the LL equation, not the LLG equation.

We thank the Reviewer for pointing out this inconsistency. It in fact should be LL, we have revised the manuscript.

The authors emphasized several times that the chaos involved in the switching is low-dimensional. How low? Can the authors estimate the fractal dimension of the chaos?

Indeed, the expression 'low-dimensional' refers to the dimension of the state space of the dynamical system. On page 1 column 1, we state "Remarkably, chaos may already arise in dynamical systems with a few degrees of freedom, such as systems described by three state variables or by two state variables in the presence of a time-varying external excitation [Perko2001]. These low-dimensional dynamical systems are particularly important for studies of chaos because time evolution of all state variables can be traceable in both experiments and numerical simulations performed for such testbed systems [Moon2004]."

In the macrospin approximation, the magnetization can be described by two space variables, for example m_z and ϕ as in Fig. 3. Therefore chaotic dynamics for this system are low-dimensional. On page 2 column 2, where we describe that the magnetization exists on the unit sphere have added the sentence "The single-domain free layer magnetization can therefore be described by two state variables, for example z-component of m and azimuthal angle ϕ in the xy plane."

We have mentioned in the introduction (page 1, col 2) that chaos can be excited also in magnetic systems with continuous degrees of freedom, leading to spin-wave turbulence. We are aware that, when a chaotic attractor is present in the dynamics, it may have fractional dimension. For such dynamics, there are special invariant sets termed stable and unstable manifold (described on pages 4 and 5). Intersections of these manifolds define regions of the state space of the stroboscopic map called 'lobes', which may have either capturing or escaping character. Escaping and capturing lobes finely intersect each other giving rise to a fractal boundary between the points remaining in the well and points escaping from the well [Ott]. This boundary arises in a way that is similar to the construction of the third middle Cantor set [Ott], which has fractional dimension $\ln(2)/\ln(3)=0.631$. In the revised manuscript, we refer the reader to a detailed discussion of this topic in Y-C Lai, T. Tél, Transient Chaos, Springer, New York, NY, 2011, which is referenced on page 7, col 2.

The authors mentioned several times that the CoFeB thin film is superparamagnetic, but for me the film is pretty much the same as the free layers in typical MTJ devices. Why is it superparamagnetic? Did the authors carry out field-cooled and zero-field-cooled measurements to determine the blocking temperature?

The Reviewer is correct that our CoFeB film is similar to those used in typical MTJ devices. We note that our unpatterned CoFeB film is not superparamagnetic, which means that the CoFeB material itself is not superparamagnetic. The free layer of our MTJ device becomes superparamagnetic only when the film is patterned into a nanodevice, in which the free layer becomes a nanoparticle. Since magnetization of our free layer lies in the sample plane, its magnetic anisotropy barrier arises from magnetic shape anisotropy (controlled by the aspect ratio of the in-plane long and short axes of the nanoparticle). We intentionally choose this aspect ratio to be not too large so that magnetic anisotropy is small enough to induce superparamagnetism in the free layer nanoparticle. In MTJ devices serving as magnetic memory elements, magnetic anisotropy is engineered to be significantly higher via control of the free layer perpendicular anisotropy, thickness and/or shape. The lowered anisotropy barrier in our MTJ devices allows for thermally activated random switching between $\pm x$ directions of magnetization on the time scale of the measurement, and thus the element is superparamagnetic, as we directly observe in our time domain measurements shown in Fig. 2b. We did not perform field- and zero-field-cooled measurements to determine blocking temperature because our microwave probes are not compatible with low temperature measurements. However, the room-temperature time domain data in Fig. 2b firmly establish the superparamagnetic nature of the free layer nanoparticle. We added an explanation of why superparamagnetism arises in our MTJ to page 3 column 1 of the revised manuscript.

On page 2, the authors wrote “microwave-assisted magnetic recording (MAMR) relies on thermally activated switching of nanoscale ferromagnets.” This may not be an accurate statement, because MAMR mainly utilizes an external microwave to reduce the energy barrier and an external static field to realize deterministic switching.

We agree that MAMR primarily uses the external microwave to effectively reduce the energy barrier and also uses an external field to induce switching. However, thermal fluctuations further assist the switching process, as is always the case for any nanomagnet at a non-zero temperature. It is, of course, possible to apply such a high magnetic field (e.g. exceeding the anisotropy field) that completely destroys one of the energy wells of a uniaxial nanomagnet and makes the switching completely deterministic. However, this is costly in practice and a smaller field can be used to induce thermally-assisted probabilistic switching with a high degree of fidelity. We extended the discussion on page 6 to stress that microwaves and external bias field are the key ingredients in MAMR while thermal assistance is employed to improve energy efficiency of this process.

On page 6, the authors wrote “deterministic chaos induced by a sub-FMR-frequency drive may be a more energy-efficient approach to energy-assisted switching of magnetization than MAMR at the FMR frequency.” I am wondering how the frequency of the magnetization precession during the switching is compared to the microwave frequency? In case they are close, the film may also take energy from the microwave, as in MAMR.

We agree with the Reviewer that the free layer nanomagnet does take energy from the the microwave drive when the drive frequency is below the linear FMR frequency. However, it is not always possible to define the precession frequency/period during the switching process. This is because a sufficiently strong microwave drive induces a-periodic motion of magnetization leading to chaos. Absorption of energy by magnetization from the microwave drive is an integral part of the ac-driven heteroclinic tangle theory and chaos-induced switching. This energy is needed to induce chaotic dynamics and erosion of the basins of stability. We find that the good agreement of the heteroclinic

tangle theory with our switching data validates this approach. On page 4 of the revised manuscript, we mention that the microwave spin torque drive can transfer energy to magnetization.

Does microwave heating play any roles in the presented switching experiments, especially in the regime where high-power microwaves were used (Fig. 2c)?

The Reviewer is right that ohmic heating by the microwave drive does take place. However, this heating cannot explain our data in Figs. 2b and 2c. Indeed, ohmic heating is independent of the frequency of applied microwave current while the threshold voltage in Fig. 2 shows clear dependence on the frequency as expected from chaos-induced effective barrier reduction. We thus conclude that ohmic heating, although present, does not mask the physics of chaos-induced switching of magnetization studied in our paper. On page 6 of the revised manuscript, we have added a discussion on the effects of ohmic heating.

In Fig. 5, the parameter T is not defined. Change “withing” to “within”.

We have added to the caption the definition of T (temperature) and fixed the typo.

Reviewer #3 (Remarks to the Author):

The results presented in the manuscript entitled "Magnetization reversal driven by low dimensional chaos in a nanoscale ferromagnet" might, indeed, be essential for the development of the emerging spintronic-based stochastic and reservoir computing devices. Furthermore, if communicated in an accessible way, the interpretation of the observe stochastic phenomena should appeal to the experts from other research fields and general audience. Unfortunately, the way the manuscript is written now does not easily allow to grasp the essence of the observed effect, in particular for the non-expert in the chaotic dynamics. I believe that the interpretation based on the stochastic resonance phenomena (which some of the co-authors are experts in) would be less complicated, and, thus, more appealing to prospective readers of Nature Nanotechnology. Stochastic resonance has been observed and understood in spintronic devices before, including the papers co-authored by the authors of the present manuscript that raises a question about the novelty of the results. So in the revised manuscript, it should be clearly stated how present results advance understanding of stochastic resonance in spintronic devices.

We thank the Reviewer for proposing an interesting idea of possibly explaining the observed phenomena in terms of stochastic resonance. We seriously looked into this possibility and performed measurements to characterize stochastic resonance driven by spin torque in our MTJ devices. These experiments are described in Supplementary Note 3 added to the manuscript. We indeed observe clear signatures of stochastic resonance. The data reveal that stochastic resonance mechanism dominates the switching process for ac current frequencies below the Kramers transition rate (sub-kHz for our samples) and becomes negligible at much higher drive frequencies discussed in the main text of the paper. We thus conclude that physics different from stochastic resonance must be invoked to explain our data at GHz frequencies shown in Fig. 2 and Fig. 4. We also note that non-adiabatic (high-frequency) stochastic resonance is not seen in our experiments as it requires application of a significant out-of-plane magnetic field to the MTJ (X. Cheng et al., Phys. Rev. Lett. **105**, 047202 (2010)). We revised the manuscript to include details of the low-frequency stochastic resonance regime on page 3 column 2 as well as a reference to the new Supplementary Note 3 for more details. This revision significantly improves the overall quality of our manuscript for two reasons: (i) the paper now covers the effect of spin torque drive on the switching process over the entire range of frequencies from dc to the FMR frequency and (ii) the manuscript gives a more convincing explanation of the observed effects via considering a wider range of mechanisms. It is interesting that we find different mechanisms to dominate over different frequency ranges with stochastic resonance being

dominant in the low frequency sector and low-dimensional chaos dominating sub-FMR GHz-scale frequencies. The latter mechanism is confirmed via comparison of the threshold voltage data in Fig 2c to analytical (Fig. 2c) and numerical (Fig. 4b) calculations. In the range of frequencies where low dimensional chaos is the only relevant mechanism (1 GHz - 4 GHz), both analytical and numerical calculations describe the experimental data well. This agreement lends strong support to the low dimensional chaos mechanism. This mechanism is indeed novel and has not been previously reported in any experiment in a magnetic system.

We also agree with the Reviewer that superparamagnetic MTJs studied here are promising building blocks for computing devices based on novel paradigms such as reservoir computing, and therefore the interplay between stochasticity and chaos is interesting from a technological viewpoint. We stress this point in the revised manuscript and include additional references on reservoir computing based on magnetic building blocks in the introduction (page 2, column 1) and concluding paragraph.

Following Reviewer's suggestions, we made revisions of the presentation to make the results of the manuscript more accessible to general audience. Since the simpler concept of stochastic resonance does not explain our data, we must use more the complicated concepts of chaotic dynamics, mathematical details of which are usually not known by the general audience. However, many questions addressed by the chaos theory are of great interest to the general public. One example is stability of the Solar System - the question that fascinated the general public for centuries and can only be answered within the framework of the chaos theory. Quite interestingly, the type of transient (switching) chaotic dynamics studied in our work belongs to the same class of problems as the Solar System stability problem. We therefore, modified the introduction and the discussion (pg. 7-8) of the manuscript to clearly connect the problem studied in our work to chaotic dynamics problems of general interest. We believe that our experimental discovery of transient chaos in a magnetic system relevant to multiple novel computing and data storage applications will be of general interest when discussed in the context of other famous problems of chaotic dynamics. This discovery adds to the unified scientific picture of reality by linking such seemingly disparate phenomena as planetary system stability and magnetic data storage in a fascinating and unexpected way. We thank the Reviewer for pointing out the importance of presentation for the general audience - we believe we significantly improved this aspect of the manuscript.

The second major remark is regarding the theory developed in the manuscript. It appears to me (Fig. 2c) that experimentally measured V_{ac}^0 vs. frequency (f) shows some \sqrt{f} behavior in the entire range of frequencies, while theory predicts f^p , where $p > 1$. So, most likely, the macrospin theory cannot grasp the underlying stochastic process. Therefore I would like authors to discuss this discrepancy and, at least, mention its possible causes.

Indeed, our zero-temperature analytical theory predicts a nearly quadratic dependence of the threshold voltage on frequency as shown in Fig. 2c. We checked the degree of validity of this theory at a non-zero temperature via numerically solving macrospin LL equation for our system at $T = 300$ K. The results of these numerical simulations are shown in Fig. 4b. This figure allows us to make two important observations: (i) finite-temperature solution of the macrospin LL equation describes our experimentally measured frequency dependence of the critical voltage over the entire frequency range of the measurement (ii) zero-temperature analytical solution for critical voltage versus frequency closely follows the numerical solution at $T=300$ K for frequencies from 1 GHz to near the FMR frequency.

The latter observation reveals that ac-induced chaos is the dominant factor impacting switching dynamics in the frequency range from 1 GHz to near the FMR frequency. Our analytical expression

predicting nearly quadratic (plus a constant) dependence of the threshold voltage on frequency is a good approximation in this frequency range.

The former observation demonstrates that a deviation from the nearly quadratic dependence at frequencies below 1 GHz is a thermal effect. This deviation can be explained by noting that negative (positive) current applied to the MTJ acts as negative (positive) damping and thus enhances (suppresses) the amplitude of thermal fluctuations of magnetization within the potential well [Petit et al., Phys. Rev. Lett. 98, 077203 (2007)]. For a sufficiently low frequency of the ac drive, the system is subjected to the negative damping effect of spin torque for a sufficiently long time (throughout the negative-damping half-period of the ac drive) so that there is a non-negligible chance that the enhanced amplitude of thermal fluctuations drives magnetization over the barrier and thereby leads to switching.

The Reviewer is correct that the measured (and numerically simulated at $T = 300$ K) dependence of the threshold voltage versus frequency may be approximated by an empirical power-law (plus a constant). However, the simulations reveal that this dependence is a result of three different mechanisms dominating over three different frequency ranges, and thus assigning a particular physical significance to the exponent of such an approximate dependence is problematic. The discussion part of the revised manuscript states: "The deviations of the threshold voltage from the value predicted by the heteroclinic tangle theory at frequencies below 1 GHz (region labeled Neel-Brown in Fig.2c) arise from adiabatic enhancement of the amplitude of thermal fluctuations of magnetization by spin torque. When the ac spin torque frequency is lower than the Neel-Brown attempt frequency for thermally activated switching [Brown1963,Suh2008], antidamping spin torque can significantly amplify the amplitude of thermally-induced magnetization precession in the half-cycle of the ac drive when spin torque acts as antidamping [Petit2007] and thereby induce switching over the energy barrier. This mechanism of thermally assisted switching is distinctly different from the heteroclinic tangle mechanism and it can significantly decrease the low-frequency value of the ac threshold voltage below that predicted by Eq.(4). To verify the role of this mechanism, we numerically solved [Daquino2006] the stochastic LL equation (1) at $T = 300$ K (Methods). The results of these simulations shown in Fig.4b are in a remarkably good agreement with the experimental data shown in Fig.4a over the entire frequency range employed in the experiment. These simulations confirm that the threshold voltage is significantly reduced by the antidamping action of ac spin torque at frequencies below the attempt frequency but is nearly insensitive to temperature at frequencies exceeding the attempt frequency."

Finally, there are at least three relatively slow processes that might explain enhanced switching rates at low injection frequencies:

- time needed for the phase oscillator to return on the limiting cycle after perturbation $1/\Gamma_p$, where Γ_p is amplitude-to-frequency conversion rate, which is typically below 1 GHz.
- (super)paramagnetic resonance frequency, $\omega = \gamma B = 0.11$ GHz for the parameters mentioned in the manuscript.
- Kramers rate that according to Fig.2a is around 100 Hz. I would like authors to discuss possible contributions of these effects to the observed results.

Overall, although I believe the manuscript might be suitable for the publication in Nature Communications, I cannot recommend it for publication in its present form.

We thank the Reviewer for pointing out the need to broaden the discussion and consider other mechanism of magnetic oscillations and relaxation in the context of the observed enhanced switching rates at sub-FMR injection frequencies. Here is our brief analysis of the three processes mentioned by the Reviewer:

1. Time needed for the phase oscillator to return on the limiting cycle after perturbation, $1/\Gamma_p$. The frequency scale due to this relaxation process, Γ_p , is indeed below 1 GHz. This time scale is only relevant for the auto-oscillatory regime of magnetization dynamics (in e.g. spin torque oscillators above the critical current). We note that the applied ac current in our experiments only exceed the critical current for a fraction of the applied ac drive period, and only for high values of power used in our measurements. Furthermore, Γ_p depends on the applied current and thus does not assume a single numerical value but constantly changes in time as the applied current oscillates. Given that Γ_p is not a constant but a complicated function of time in our experiments, it is hard to utilize this time-dependent rate for a meaningful analysis of our data.

2. (Super)paramagnetic resonance at frequency $\omega = \gamma B = 0.11$ GHz, where B is the applied field. This is precession frequency of electron spin in applied magnetic field B when magnetic anisotropy fields are much smaller than B. Since magnetic anisotropy field of our superparamagnetic nanoparticle (MTJ free layer) is much higher than the applied external field, this frequency $\omega = \gamma B = 0.11$ GHz is not relevant for our system. The frequency of 0.11 GHz would be relevant if we had free paramagnetic (not exchange coupled to each other) electron spins in the system, which we do not expect to be present and see no evidence of being present in our MTJs.

3. Kramers rate near 100 Hz. This Kramers transition rate is indeed relevant for stochastic resonance phenomena that we observe at low frequencies ($\ll 1$ MHz). We describe the relevance of this transition rate in the new Supplementary Note 3. This Kramers rate, however, is not relevant to the GHz-frequency-scale enhancement of switching rates in Fig. 2 and Fig. 4.

REVIEWERS' COMMENTS:

Reviewer #2 (Remarks to the Author):

Since the authors have sufficiently addressed all of my questions and the manuscript has been significantly improved, I would like to recommend the publication of this paper in Nature Communications. I would think this paper should be of great interest to the magnetic recording and magnetic memory communities, but the in-plane magnetization configuration (not perpendicular) and the superparamagnetic regime (low thermal stability) are not favored/desired in terms of practical applications. Some discussions about this in the end of the paper should be very useful.

Reviewer #3 (Remarks to the Author):

I would like to thank authors of the manuscript for thoughtfully addressing my comments, suggestions and questions. Overall, I still believe the manuscript cannot be easily followed by the non-experts in the field of chaotic and nonlinear magnetization dynamics. Although manuscript might catch prospective readers attention by referring to many similar phenomena in other research fields and applications in emerging STT-MRAM technology, it is particularly hard to follow what is the chaotic magnetization dynamics and why it happens in the MTJs.

Also, I would disagree with authors that Γ_p is not relevant to their results. In a broad sense, it describes relaxation rate of the amplitude fluctuations. As it was experimentally verified in <https://aip.scitation.org/doi/10.1063/1.4898093>, it defines the bandwidth of the amplitude modulation via ac damping-like torque. One would expect, the same limitation applies to the chaotic dynamics.

Anyway, I tend to believe that in its present form the manuscript is more suitable for more specialized, but high impact journal such as PRL.

We would like to thank again all the Reviewers for their time in evaluating and commenting on our manuscript. Please find below our point-by-point replies to the Reviewer comments. Our replies (indented blue text) are interlaced with the Reviewers' original comments. We have made modifications to the manuscript for improved clarity as detailed below.

REVIEWERS' COMMENTS:

Reviewer #2 (Remarks to the Author):

Since the authors have sufficiently addressed all of my questions and the manuscript has been significantly improved, I would like to recommend the publication of this paper in Nature Communications. I would think this paper should be of great interest to the magnetic recording and magnetic memory communities, but the in-plane magnetization configuration (not perpendicular) and the superparamagnetic regime (low thermal stability) are not favored/desired in terms of practical applications. Some discussions about this in the end of the paper should be very useful.

We thank the Reviewer for positive evaluation of our revised manuscript and recognizing the potential importance of the reported effect to the magnetic recording and memory technologies. We agree that memory applications require high thermal stability. To address this concern, we have added a note in the final paragraph of the revised manuscript: "While we have demonstrated the effect in nanoscale magnetic tunnel junctions with superparamagnetic free layers, ac-driven chaos is also expected to facilitate switching of thermally-stable free layers employed in non-volatile memory applications." The choice of superparamagnetic free layer MTJs in our work was for the purpose of accelerated testing. Additionally, as mentioned in the manuscript, MTJs are also attractive for neuromorphic and reservoir computing, where the superparamagnetic regime is actually desirable [Locatelli, N., Cros, V. & Grollier, J. Spin-torque building blocks. *Nat. Mater.* 13, 11–20 (2013), Locatelli, N. et al. Noise-enhanced synchronization of stochastic magnetic oscillators. *Phys. Rev. Appl.* 2, 034009 (2014), Mizrahi, A. et al. Neural-like computing with populations of superparamagnetic basis functions. *Nat. Commun.* 9, 1533 (2018)].

Reviewer #3 (Remarks to the Author):

I would like to thank authors of the manuscript for thoughtfully addressing my comments, suggestions and questions. Overall, I still believe the manuscript cannot be easily followed by the non-experts in the field of chaotic and nonlinear magnetization dynamics. Although manuscript might catch prospective readers attention by referring to many similar phenomena in other research fields and applications in emerging STT-MRAM technology, it is particularly hard to follow what is the chaotic magnetization dynamics and why it happens in the MTJs. Also, I would disagree with authors that Γ_p is not relevant to their results. In a broad sense, it describes relaxation rate of the amplitude fluctuations. As it was experimentally verified in <https://aip.scitation.org/doi/10.1063/1.4898093>, it defines the bandwidth of the amplitude modulation via ac damping-like torque. One would expect, the same limitation applies to the chaotic dynamics. Anyway, I tend to believe that in its present form the manuscript is more suitable for more specialized, but high impact journal such as PRL.

We thank the Reviewer for carefully reading our revised manuscript. We are pleased the Reviewer believes our work is suitable for publication in a high impact journal such as PRL. However, we feel that publishing in a specialized journal would miss several important target groups for this work, including magnetic recording community, neuromorphic computing community and a large fraction of nonlinear dynamics community. We believe Nature Communications is the perfect journal for publication of our work relevant to multiple science and engineering communities.

We agree with the Reviewer that the relaxation rate $\Gamma_p < 1$ GHz is relevant because it defines the bandwidth of the amplitude modulation via ac damping-like torque as shown in [M. Quinsat et al., *APL* 105, 152401 (2014) <https://aip.scitation.org/doi/10.1063/1.4898093>]. As such, this relaxation rate impacts ac-driven dynamics at frequencies below ~ 1 GHz and is not relevant for ac-driven dynamics at frequencies

above ~ 1 GHz. As we show in present work, the switching dynamics for ac drive frequencies above 1 GHz is dominated by deterministic chaos (see Fig. 2c) and can be described by our analytical expression (Eq. 4) without invoking Γ_p , as expected. We describe the switching dynamics below 1 GHz via numerically solving stochastic LL equation (Eq. 1), which automatically includes all relevant physics, including that of the amplitude relaxation described by Γ_p . To clarify this point, we add the following sentence to the first paragraph on page 6 of the revised manuscript: "It has been previously shown that efficient amplification of the free layer magnetization amplitude by ac spin torque drive is limited to frequencies below the free layer magnetization relaxation rate, which typically does not exceed 1 GHz [M. Quinsat et al]."